# Metal Halide Perovskite Nanowires: Controllable Synthesis, Mechanism, and Application in Optoelectronic Devices

**DOI:** 10.3390/nano13030419

**Published:** 2023-01-19

**Authors:** Yangbin Lu, Kang Qu, Tao Zhang, Qingquan He, Jun Pan

**Affiliations:** College of Materials Science and Engineering, Zhejiang University of Technology, Hangzhou 310014, China

**Keywords:** perovskite, nanowires, photodetector, laser, solar cells

## Abstract

Metal halide perovskites are promising energy materials because of their high absorption coefficients, long carrier lifetimes, strong photoluminescence, and low cost. Low-dimensional halide perovskites, especially one-dimensional (1D) halide perovskite nanowires (NWs), have become a hot research topic in optoelectronics owing to their excellent optoelectronic properties. Herein, we review the synthetic strategies and mechanisms of halide perovskite NWs in recent years, such as hot injection, vapor phase growth, selfassembly, and solvothermal synthesis. Furthermore, we summarize their applications in optoelectronics, including lasers, photodetectors, and solar cells. Finally, we propose possible perspectives for the development of halide perovskite NWs.

## 1. Introduction

Metal halide perovskites have gained significant attention for various optoelectronic applications, such as light-emitting diodes, [1,2,3,4,5] lasers, [6,7,8] photodetectors (PDs), [9,10,11] and solar cells [12,13]. Recently, researchers have shown increasing interest in perovskite materials with different structures, such as quantum dots, [14,15] nanowires (NWs), [16,17,18,19,20,21,22] nanoplates, [23,24] and thin films [25,26,27]. One-dimensional (1D) perovskite NWs have been extensively studied because of their anisotropic properties and quantum mechanical effects. In addition, NWs have been used as building blocks in various applications, such as electronics, optoelectronics, sensing, and nanoscale energy harvesting [28]. Furthermore, owing to their excellent crystallinity and controllable interface engineering, single-perovskite NWs or their assemblies are ideal models for investigating charge carrier dynamics [29]. 

Compared with their thin films or bulk counterparts, perovskite NWs have many advantages, including negligible ionic defects and grain boundaries, and thus exhibit enhanced photogenerated carrier transport properties, which can significantly improve the performance, reliability, and stability of optoelectronic devices [18,20].Furthermore, the inherently large surface area of perovskite NWs significantly enhances their light-harvesting properties. The spatial confinement of charge carriers in the highly crystalline 1D structure further improves charge separation, extraction, and transport [30]. These advantages make 1D perovskite NWs promising materials for next-generation photovoltaic applications. This review summarizes recent synthetic strategies for perovskite NWs, such as the hot injection (HI), vapor-phase, and selfassembly methods. Their applications in the optoelectronic fields (lasers, PDs, and solar cells) are also listed. Finally, we highlight the current problems and possible perspectives of perovskite NWs, and we hope that this review will contribute to further developments in the perovskite field.

## 2. Synthesis of Perovskite Nanowires

A typical perovskite chemical formula is ABX_3_, as shown in Figure 1a,b, where A is an inorganic or organic cation, such as Cs^+^, CH_3_NH^3+^ (MA^+^) or NH_2_CHNH^2+^ (FA^+^), B is the divalent cation Pb^2+^ or Sn^2+^, and X is a halogen anion (Cl^−^, Br^−^ or I^−^) [31,32,33]. Unlike ordinary semiconducting nanocrystals (NCs), halide perovskite NCs have fragile ionic bonding with dynamic capping ligands, which leads to further growth of NCs along low-energy directions to form 1D NWs [34,35]. Most perovskite NWs are synthesized by solution processes, such as hot injection, selfassembly, solvothermal, and anion exchange, exhibiting the advantages of high yield, aspect ratio regulation, large area, and easy transfer to the device. Vapor-phase growth provides high-quality crystals and low defect density, thus effectively improving the performance of optoelectronic devices [36]. In this section, we review the synthetic strategies of perovskite NWs in recent years, including the hot injection method, vapor phase synthesis method, and selfassembly induced by different factors. The morphological differences between NWs synthesized using each method are summarized in Table 1.

### 2.1. Hot Injection

The hot injection (HI) method has been previously developed to synthesize cadmium chalcogenides. The organometallic reagent was rapidly injected into a thermally co-ordinated solvent to generate discrete uniform nucleation [55]. Hot injection leads to instantaneous nucleation, quenched by the rapid cooling of the reaction mixture, as supersaturation is relieved by a nucleation explosion. At lower temperatures, this causes further growth of the existing nuclei into mature NCs without causing new nucleation [56]. In recent years, the HI method has been used to synthesize perovskite NCs. Protesescu et al. fabricated monodisperse CsPbX_3_ NCs with cubic shape for the first time by the HI method [37]. Typically, Cs_2_CO_3_ was added to a mixture of octadecene (ODE) and oleic acid (OA) and heated under nitrogen to form a Cs precursor. Subsequently, the cesium precursor was injected into a mixture of PbX_2_, ODE, oleylamine (OAm), and OA at high temperatures (150–200 °C) (Figure 1c). The reaction system was then quenched in an ice bath to obtain NCs of different sizes, which could be adjusted by controlling the reaction conditions. Notably, the obtained NCs with different compositions displayed tunable emission spectra from 410 nm to 700 nm (Figure 1d) [29,37]. 

Interestingly, high temperatures and long reaction times could promote the generation of NWs. Yang et al. investigated the reaction kinetics of CsPbBr_3_ NW synthesis at 150 °C. In the initial stage (t < 10 min), the reaction was dominated by the formation of nanocubes (Figure 2a). After 10 min, thin nanoribbons with diameters of approximately 9 nm were observed (Figure 2b). As time increased, the number of NCs decreased, and some square nanosheets were formed (Figure 2c). At the later stage of the reaction (40–60 min), the nanosheets dissolved, and the main products were NWs with uniform diameters and lengths of up to 5 μm (Figure 2d). However, the NWs disappeared over time, and the final products were large crystals (Figure 2f) [16].

Among all-inorganic halide perovskites, cubic-phase (α) CsPbI_3_ has the narrowest band gap and shows great potential for use in many optoelectronic devices. Chen et al. proposed a new two-injection method at 60 °C and 120 °C to control the growth of α-CsPbI_3_ NWs, as shown in Figure 3a–d. α-CsPbI_3_ NWs were synthesized via the reaction of cesium oleate with lead halides in a mixture of ODE, OA, and OAm. In the experiment, a low synthesis temperature of 60 °C was used to dissolve PbI_2_. Two injections of the Cs precursor and an extended 30 min reaction time per injection allowed the NWs to grow fully under the protection of the ligand. Stable α-CsPbI_3_ NWs were synthesized using this method and stored in an inert atmosphere for more than three months [38].

Generally, long-chain alkyl carboxylic acids (OA) and alkyl amines (OAm) are used as ligands to synthesize perovskite NWs. Manna’s group fabricated different, thin CsPbBr_3_ NWs using a modified HI method by tuning the concentration of short alkyl carboxylic acids (octanoic acid or hexanoic acid) and alkyl amines (octylamine and OAm). As shown in Figure 3e, the widths of the thin NWs are 10, 5.1, and 3.4 nm, respectively [39]. Semiconductor NWs with diameters smaller than the exciton Bohr radius have received considerable attention owing to their unique optical properties. Yang et al. developed highly uniform ultrathin CsPbBr_3_ NWs with a diameter of 2.2 ± 0.2 nm, much lower than the exciton Bohr radius of CsPbBr_3_ (Figure 3f). After purification and surface treatment, the ultrathin NWs exhibited bright emissions with a PL quantum yield (PLQY) of approximately 30% [40].

### 2.2. Vapor Growth

Vapor-phase growth converts a crystalline material into a vapor phase through sublimation, evaporation, sputtering, or decomposition and then deposits it under appropriate conditions to realize the controllable atomic transfer of substances from the source material to the solid film [57]. When compared to hot injection methods, the advantage of vapor-phase growth techniques is that the grown NWs tend to be of higher quality and have lower defect density. Vapor growth is typically performed in chemical vapor deposition (CVD) tubular reactors, which provide a controlled and scalable method for growing high-quality semiconductors [58,59]. CVD has been widely used to fabricate halide perovskite NWs with excellent properties [41,60,61]. Xing et al. prepared PbI_2_ NWs synthesized on silicon oxide substrates by the CVD method and then converted them to MAPbI_3_ NWs by a gas-solid heterophase reaction with MAI molecules. The original geometry of the NWs was maintained after the conversion (Figure 4a–d). Furthermore, MAPbBr_3_, MAPbI_x_Cl_3–x_, MAPbI_x_Cl_3–x_, MAPbIxBr_3–x_, and MAPbBr_x_Cl_3–x_ NWs were prepared using the same method [41]. 

Black cubic-phase CsPbI_3_ has excellent photoelectric properties; however, it prefers phase transition at room temperature, thus limiting the development of devices [31].Fan et al. fabricated high-density vertical arrays of stable CsPbI_3_ NWs in anodic aluminum membranes (AAMs). CsPbI_3_ NWs were grown from lead metal nanocluster seeds using a simple vapor-phase growth method (Figure 4e), and the NW arrays grown in AAMs had a stable cubic phase with excellent optical properties [42]. The vapor-phase method can grow high-quality epitaxial nanostructures of inorganic perovskites owing to their thermal stability at high temperatures [62,63]. Song et al. grew high-quality inorganic CsPbX_3_ NWs and microwires with horizontal orientation on mica via vapor-phase epitaxy. By varying the growth time, single NWs, Y-shaped branches, and interconnected NW networks with six-fold symmetry were obtained (Figure 4f). These well-connected CsPbBr_3_ NWs showed great potential for optoelectronic applications [43].. In addition to mica substrates, sapphire substrates can also be used to fabricate inorganic perovskite NWs. Joselevich et al. explored uniform CsPbBr_3_ NWs with horizontal and directional orientations on flat and polyhedral sapphire surfaces, forming six-fold symmetric and two-fold symmetric arrays along specific directions of the sapphire substrates, respectively. They found a size-dependent PL emission shift well beyond the quantum range, and the emission peak redshifted from 514 nm to 528 nm as the diameter of the NWs changed from 45 nm to 1.5 μm (Figure 4g,h) [44].

Conventional solution- and vapor-phase methods are used to grow CsPbX_3_ NCs through a noncatalytic and straightforward vapor-solid (VS) synthesis process rather than a catalytic vapor-liquid-solid (VLS) growth mechanism. However, the nanostructures obtained via the noncatalytic VS growth process are poor, particularly the uncontrolled radial growth along the sidewall of the NWs [44]. The VLS mechanism represents an advantage of NW synthesis, which can accurately regulate the size, position, and surface characteristics of the NWs [[64],[65],[66],[67],[68],[69]，[70]]. 

As shown in Figure 4i, Cahoon et al. first fabricated PbI_2_ NWs autocatalytically using the VLS process and then converted the NWs to MAPbI_3_ NWs via the low-temperature vapor-phase intercalation of MAI [45]. Similarly, Shin et al. used a PbI_2_ film to obtain highly uniform and dense arrays of oriented PbI_2_ NWs by VLS growth and then transformed them into MAPbI_3_ NWs through MAI in the vapor phase [46]. In addition to organic-inorganic perovskite NWs, VLS growth has also been applied to fabricate all-inorganic perovskite NWs. As shown in Figure 4j, Meng et al. developed a VLS growth method for a large vertical CsPbX_3_ NW array with a uniform diameter. The Sn catalyst in the liquid state and supersaturated Cs, Pb, and X precursors induced the growth of CsPbX_3_ NWs at the liquid-solid interface [47].

### 2.3. Selfassembly

Selfassembly is a process of arranging individual components into ordered structures [34]. Perovskite NCs exhibit more unstable ligand dynamics and defects than conventional semiconductor NCs do [71,72,73]. Improving device performance by selfassembling NCs into NWs for surface defect passivation has been widely studied [32,74].Various external stimuli, such as solvent, temperature, and light, affect the selfassembly of NCs [35,75].

#### 2.3.1. Ligand-Assisted

Perovskite quantum dots (PQDs) have soft lattices, low formation energies, unstable ligand dynamics, and more common defects than those of conventional semiconductor QDs [71,72]. Defects are critical for controlling the electrical and optical properties of perovskite materials [76]. In addition, the presence of defects leads to poor storage stability. The passivation of surface defects can enhance the stability and PLQYs of luminescent nanoparticles. Zhong et al. applied a ligand-assisted reprecipitation method to synthesize MAPbX_3_ QDs at room temperature [77]. MAPbBr_3_ NWs were fabricated by controlling the number of surfactant ligands [48]. Compared with the selfassembly of organic-inorganic perovskite NWs, Pan et al. synthesized CsPbBr_3_ PQDs (halide-vacancy-rich PQDs, HVPQDs) using a traditional HI method. The synthesized PQDs were selfassembled by the addition of OA and dodecyldimethylammonium sulfide (DDAS). As shown in Figure 5a–d, the morphologies of the precipitates formed at different stages were observed by HRTEM. At the initial stage (t = 5 min), the particle size of the HVPQDs increased (Figure 5b). The nanoparticles were fused in one direction, and a NW shape was observed at t = 60 min (Figure 5c). Over time, the nanoparticles were transformed into NWs of approximately 20–60 nm in width and lengths of a few millimeters (Figure 5d). The proposed selfassembly mechanism is illustrated in Figure 5e. Additional OA alters the surface ion equilibrium of the HVPQDs, and the S^2−^ group from DDAS occupies Br vacancies. Then, two adjacent HVPQD monomers prefer to attract owing to divalent sulfur, thereby inducing the selfassembly process [49].

#### 2.3.2. Light-Induced

Light can be delivered immediately to a precise location and closed system [75]. Halide perovskites tend to change in response to light, such as photoinduction, which affects their phase transformation, nucleation growth, and ion migration [78,79,80,81]. Light-induced synthesis and controlled photoinduced selfassembly are promising approaches for constructing novel structures and materials with potential applications in optical, sensing, and transmission systems [82]. Liu et al. prepared defect-free NWs by assembling and fusing single cubic CsPbBr_3_ NCs under visible light irradiation (Figure 6a). After photoexcitation, the dissociated charged excitons diffuse to the perovskite NC surface and are trapped by the surface ligands. Light irradiation efficiently detached the surface ligands, and the NCs fused via bare surface contact. In order to elucidate the NW formation mechanism, the growth processes at different stages were monitored by HRTEM. During the initial phase (0–2 h), the NCs tended to align to form a linear structure (Figure 6b). Subsequently, nanorods appeared after approximately 2–5 h (Figure 6c). The NCs gradually disappeared over time, and the nanorods were continuously grafted into NWs (Figure 6d). Adjacent NWs were attached and subsequently transformed into nanoribbons, even as longer and thicker NWs, as shown in Figure 6e. The final product was a mixture of nanoribbons and NWs with diameters of hundreds of nanometers and chain lengths of 10–20 μm. They formed super-long but flexible chains of over 50 μm, equivalent to approximately 7000 tightly packed NCs [50].

#### 2.3.3. Polar Solvent Induction

Polar solvents greatly influence halide perovskites in terms of their luminescence properties, morphological characteristics, and surface defects [83,84]. Sun et al. reported that polar solvent molecules can induce lattice distortion in CsPbI_3_, triggering dipole moments and leading to the transformation of CsPbI_3_ nanocubes into single-crystal NWs. As shown in Figure 7a, the twisted nanocubes selfassembled and recrystallized into single-crystal NWs owing to the dipole–dipole force. In order to minimize the surface energy, the assembled NWs can be further selfassembled side-by-side into thick single-crystal NWs with diameters of up to submicrons [51]. Similarly, as shown in Figure 7b, Peng et al. obtained CsPbBr_3_ NWs by using the room-temperature supersaturated recrystallization method. The precursor in N,N-dimethylformamide (DMF) solution was mixed with toluene and acetonitrile (ACN), and CsPbBr_3_ NWs were rapidly generated with a cubic-to-orthorhombic phase transition [52].

### 2.4. Solvothermal

Although various methods have been used to prepare ultra-thin CsPbX_3_ NWs, the yields of these NWs is low [40]. In addition, stability is another issue when the diameter of the NWs is too small [39]. The solvothermal method involves the addition of a perovskite precursor and solvent to a high-pressure reaction kettle and reaction at a specific temperature for a particular time. When compared with existing synthetic methods, the solvothermal method involves simple steps, the precise control of the composition and morphology, and the high crystallinity of the synthesized products. Giri et al. added OA and OAm to a DMF solution containing MABr and PbBr_2_, and the mixture was allowed to react in an autoclave for 30 min. They regulated MAPbBr_3_ NCs with different morphologies by varying the reaction temperature (60 − 180 °C), as shown in Figure 8a [85]. Chen et al. also synthesized high-quality all-inorganic perovskite nanocrystals using a solvothermal method. As shown in Figure 8b, when the precursor was heated directly without dissolution, the concentration of the precursor ions was relatively low. The precursor dissolved gradually with increasing temperature and a relatively small amount of CsPbX_3_ NCs nucleated at a specific precursor concentration. With the help of the capping ligands, the nuclei eventually grew into nanocubes. In contrast, predissolving precursors can form a high concentration of precursor ions. More nuclei formed when the solution was heated, resulting in smaller nuclei. Finally, uniform NWs were formed by increasing the growth time instead of the nanocubes [53].

### 2.5. Anion Exchange

Ion exchange reactions on colloidal nanocrystals provide pathways for fine-tuning the composition or other inaccessible materials and forms. Cation exchange is easy and common, whereas anion-exchange reactions are relatively difficult [86]. To date, the anion exchange method has been successfully applied to perovskite NW fabrication, enabling a tunable emission range. The Kovalenko group prepared all-inorganic CsPbX_3_ NCs by adding a halide source (lead halide or alkenyl ammonium halide) using an anion exchange method (Figure 9a) [86]. Yang et al. used highly monodisperse CsPbBr_3_ NWs as a template to control the composition of NWs via an anion-exchange reaction. Various alloy compositions were obtained, and their morphologies and crystal structures were maintained (Figure 9b). The NWs exhibited high luminescence properties with PLQY ranging from 20% to 80% [28]. Moreover, as shown in Figure 9c, spatially resolved multicolor CsPbX_3_ NW heterojunctions were synthesized using the anion exchange method, and the luminescence was adjustable throughout the visible spectrum [87]. Polavarapu et al. also used synthesized CsPbBr_3_ NWs with uniform widths as templates and achieved tunable bandgaps and a PL of NWs over the entire visible light range of 400–700 nm by anion exchange (Figure 9d–h) [54]. Metal halide perovskites are prone to chemical transformation through ion exchange, which is attributed to their “soft” crystal lattice, enabling rapid ion migration [88,89].

## 3. Application of Perovskite Nanowires

One-dimensional halide perovskite NWs have become attractive candidates for next-generation optoelectronic devices because of their low defect density, long carrier lifetime, superior optical capture, and other properties [21,44,90]. In recent years, the excellent properties of halide perovskite NWs have led to their application in various optoelectronic devices (lasers, PDs, and solar cells), as summarized in Table 2.

### 3.1. Lasers

When compared to polycrystalline thin films, NWs exhibit a higher crystal mass, longer carrier diffusion length, and greater gain for lower threshold applications [41]. The superior properties of perovskite materials effectively support their potential development in lasers. Zhu et al. fabricated a perovskite NW laser by using high-quality MAPbBr_3_, MAPbI_3_, and MAPb(Cl, Br)_3_ NWs. The obtained NWs exhibit smooth end facets, making them ideal Fabry–Pérot cavities for lasing applications. The lasing from the NWs displayed a low threshold of 220 nJ cm^−2^ and a high-quality factor of 3600 at room temperature [91]. Qu et al. proposed a metal-cavity plasma nanolaser based on a single Au-MAPbBr_3_ NW prepared using a one-step in situ solution. The competition between the photonic and plasmonic modes was observed by changing the size of the hybrid plasma nanolaser (150–500 nm). The nanolasers exhibited attractive lasing behavior with a threshold of 375 μJ cm^−2^ and Q value of 589 [92]. Although MAPbX_3_ has shown excellent laser performance, instability is a major obstacle to its photoelectronic applications [115]. Jin et al. reported lead-halide perovskite single-crystal NWs (FAPbX_3_) grown in a low-temperature solution. FAPbBr_3_ NWs are optically pumped at room-temperature near-infrared (~820 nm) and green lasers (~560 nm) with low lasing thresholds of several μJ cm^−2^ and high-quality factors of 1500–2300. (FA, MA)PbI_3_ and (FA, MA)Pb(I, Br)_3_ have also been demonstrated to be tunable nanowire lasers over a wider wavelength range via cation and anion replacement (Figure 10a–e). Compared to MA-based perovskite nanomaterials, FA-based perovskite nanomaterials have better photostability and wider wavelength tunability, suggesting that FA-based perovskites may be more promising and stable candidates for new perovskite-based lasers [93].

Various mechanisms have been proposed to explain the perovskite lasers, such as the plasma mode and photon mode [15,91,116,117,118].Plasma lasers typically have lower thresholds than those of conventional photon lasers [116,119]. Jin et al. introduced a single-crystal MAPbI_3_ NW into a plasma laser as an organic-inorganic semiconductor gain material, achieving a lower threshold (13.5 μJ cm^−2^) under ambient conditions [119]. Lu et al. recently presented a 4f measurement system based on the reconstruction of the near-field of hybrid plasmonic MAPbBr_3_ perovskite nanolasers to determine the exact resonant mode [116]. The scaling laws for the wire-type hybrid plasmonic perovskite lasers were successfully determined by combining them with other optoelectronic measurements. For example, using a perovskite wire with a width less than 2.5 µm, a hybrid plasmonic perovskite laser displayed low power consumption and strong light-matter interaction with a maximum group index of 24.2, which was superior to that of a photonic laser.

When compared to organic-inorganic perovskite NW lasers, devices based on all-inorganic perovskite NWs have a wider wavelength-tunable range and excellent stability. Yang et al. reported that CsPbBr_3_ NWs were grown in the solution phase at low temperatures and exhibited a lower maser threshold and relatively high-quality factor (Q-factor). Under light excitation, Fabry–Perot lasing occurred in the CsPbBr_3_ NWs with an onset of 5 μJ cm^−2^, and the maximum quality factor of the NW cavity was 1009 ± 5 (Figure 10f–l). The NWs were stable under environmental conditions and continuous laser irradiation. Wavelength adjustability was also demonstrated via halide substitution [8]. As shown in Figure 11a–e, Jin et al. synthesized a series of all-inorganic single-crystal NWs (CsPbX_3_, CsPb(Br,Cl)_3_, and CsPb(Br,I)_3_) using a vapor-phase halide exchange method. These NWs exhibit room-temperature lasing with low thresholds, high-quality factors, excellent photostability, and tunable wavelengths across the entire visible spectrum region (420–710 nm) [17]. Single-perovskite alloy NW can emit lasers extensively and continuously, making them ideal materials for the miniaturization and integration of optoelectronic devices. Zhang et al. proposed a solid negative-ion diffusion process to prepare a single CsPbCl_3−3x_Br_3x_ perovskite alloy NW with an adjustable bandgap of 2.41–2.82 eV. A single CsPbCl_3−3x_Br_3x_ NW realizes a wide range of continuously tunable nanolasers from 480 to 525 nm [94]. Pan et al. synthesized high-quality aligned CsPbX_3_ NWs on M-plane sapphire substrates using a vapor-phase method, and the wavelength-tunable NW-based lasers exhibited low lasing thresholds and high-quality factors (Figure 11f–i) [95]. Similarly, Yan et al. directly synthesized composition-graded CsPbBr_x_I_3−x_ NWs on mica via vapor-phase epitaxial growth. Through the asynchronous deposition of cesium lead halide and temperature-controlled anion-exchange reactions (Figure 11j–k), a graded composition of NWs, with increased Br/I from the center to the end, was obtained and can simultaneously realize dual-color lasing under femtosecond laser excitation [96]. CsPbBr_3_ NWs were integrated with nanostructured indium tin oxide substrates, which exhibit near-unity refractive indices and high electrical conductivities. This approach effectively reduced the laser mode leakage, and the NWs device exhibited a high Q factor of 7860 at a low optical pumping threshold (13 μJ cm^−2^) [97].

### 3.2. Photodetectors

In addition to lasers, perovskite materials have been widely studied in photodetectors (PDs). PDs are sensors that absorb light and convert it into electrical signals. PDs are essential components in many applications, including imaging, machine vision, and display technologies [120]. The 1D halide perovskite micro/nanostructure has a high crystalline mass, large surface–volume ratio, and anisotropic geometry, which results in a low defect/trap density, long charge carrier lifetime, and reduced carrier recombination rate [121]. In addition, 1D perovskite NWs can provide a more direct charge transfer path, which is conducive to effective charge carrier collection by electrodes. Hence, PDs based on perovskite NWs can potentially exhibit excellent performance [98].

Zhou et al. reported selfpowered perovskite-NW PDs with a vertical P-I-N structure, in which CsPbBr_3_ NWs were grown using a combination of a solution-phase process and halide exchange (Figure 12a–e). The CsPbBr_3_ NW PDs show high selfsupplied electrical performance with ultrahigh on/off ratios of above 10^6^, a responsivity of up to 0.3 A W^−1^, and a detectivity of up to 1 × 10^13^ Jones [98]. Semiconductor NWs with large-scale directional growth are required for future device developments [122,123]. Pan et al. reported the successful growth of ultra-long directional CsPbBr_3_ NWs in the gas phase using the figure-axis effect on an annealed M-faced sapphire substrate. These NWs were well aligned along the substrate grooves and were several millimeters in length, enabling a high-performance PD with a responsivity of 4400 A W^−1^ and a response speed of 252 μs (Figure 12f) [99]. Owing to the toxicity of Pb in perovskite materials, their commercial development remains challenging. Nontoxic tin is considered a potential replacement for Pb. Tang’s group used a two-step solution method to synthesize CsPb_x_Sn_1−x_(Br_y_I_1−y_)_3_ perovskite NWs. As shown in Figure 12g, the PDs of the perovskite NWs exhibited high performance with a rise/fall time of 4.25/4.82 ms and a detectivity of 2 × 10^10^ Jones [100].Zhou et al. adopted the hot-pressing welding method to reduce the interface and improve the quality of the CsSnI_3_ NWs. A high-performance NW PD was obtained with a responsivity of 9.9 × 10^−3^ A W^−1^ and a detectivity of 7.2 × 10^8^ Jones [101].

In addition to all-inorganic perovskite NW PDs, PDs based on organic-inorganic perovskite NWs have also been explored by researchers. Asuo et al. deposited a perovskite NW network on a patterned substrate under environmental conditions with a relative humidity of higher than 50% using a two-step spiral coating process (Figure 13a,b). A stable hybrid perovskite NW network was synthesized under environmental conditions by directly introducing lead thiocyanate (Pb(SCN)_2_) into a precursor solution. The response time of the corresponding device was faster than 50 s under a 2 V bias, and the responsivity was improved to approximately 0.23 A W^−1^. The best PD showed a high specific detectivity of 7.1 × 10^11^ cm Hz^1/2^ W^−1^ [102]. The preparation of perovskite NWs under ambient conditions is susceptible to moisture, resulting in many grain boundaries/surface defects in the NWs. Hence, the development of an effective method is necessary to realize high-quality single-crystal perovskite NW with large-area homogeneity. Jie et al. fabricated high-quality single-crystal MAPb(I_1–x_Br_x_)_3_ (x = 0, 0.1, 0.2, 0.3, and 0.4) NW arrays on SiO_2_/Si substrates by using a facile fluid-guided antisolvent vapor-assisted crystallization method. As shown in Figure 13c–e, PD made of perovskite NW arrays exhibit a high responsivity of 12,500 A W^−1^ and a detectivity of 1.73 × 10^11^ Jones. In addition, optical detection in the wavelength range 680–780 nm can be achieved by changing the halogen ratio in the perovskite NWs [103]. Similarly, the preparation of high-quality MAPbI_3_ NWs by saturated vapor-assisted crystallization was reported by Zhang et al. [104]. Consequently, the organic solvent effectively blocked moisture entry into the NWs and enhanced their stability in the air. PDs fabricated from perovskite MAPbI_3_ NWs exhibit highly sensitive visible-light detection performance with a high responsivity of 460 A W^−1^ and a detectivity of 2.6 × 10^13^ Jones (Figure 13f–k). For optoelectronic devices, disordered NW may lead to poor device performance and low utilization of NWs [124]; hence, the uniform distribution, patterning, and alignment of the NWs is required. Song et al. achieved NW patterning and alignment through selective region processing and improved evaporation-induced selfassembly. The MAPbI_3_ NW array PDs exhibit a response time of ~ 0.3 ms, a responsivity of 1.3 A W^−1^, and a detectivity of 2.5 × 10^12^ Jones [105]. Wu et al. reported layered perovskite (MTEA)_2_(MA)_n−1_PbnI_3n+1_ single-crystal NW arrays. The sulfur–sulfur interaction between alkylammonium 2-(methylthio) ethylamine (MTEA) cations promotes crystallization by controlling the nucleation and growth of perovskites in capillary bridges, and NW arrays with single crystallinity and pure crystallographic orientation were obtained. When the crystallinity of layered perovskites (n = 3) is enhanced, ultrasensitive PDs based on NW arrays exhibit a responsivity of 7.3 × 10^3^ A W^−1^ and a specific detectivity of 3.9 × 10^15^ Jones [106]. Adding 1-butyl-3-methylimidazolium tetrafluoroborate (BMIMBF4) as an additive to MAPbI_3_ NWs improved device performance owing to fewer defects, better crystallization, enhanced light absorption, and improved charge transfer in MAPbI_3_ NWs. The NW PD exhibited remarkable performance with a detectivity, linear detection range, and noise-equivalent power of 2.06  ×  10^13^ Jones, 160 dB, and 1.38  × 10^−15^ W Hz^−1/2^, respectively, and the stability of the device was effectively improved [107].

### 3.3. Solar Cells

The excellent photoelectric properties of perovskite materials render them suitable as light-absorbing layers in solar cells. In recent years, perovskite solar cells (PSCs) have been extensively studied, and their power conversion efficiencies (PCEs) have been rapidly improved [125,126]. However, most of the light-absorbing layers of perovskite solar cells are polycrystalline perovskite films with a 3D structure, and many defects exist in the grain boundaries, deteriorating solar cell performance [127]. The excellent photoelectric properties of 1D perovskite materials make them suitable for solar cell applications. For example, low-dimensional MAPbI_3_ is expected to outperform the three-dimensional structure, improving hole migration from perovskite to the hole transport layer (HTL) and separation at the HTL/perovskite interface, Park et al. reported PSCs based on MAPbI_3_ NWs grown a using two-step spin-coating technology. The MAPbI_3_ NW-based device exhibited a short-circuit photocurrent density (J_SC_) of 19.12 mA cm^−2^, open-circuit voltage (V_OC_) of 1.052 V, fill factor (FF) of 0.721, and PCE of 14.71% [108]. Li et al. prepared perovskite MAPbI_3_ films with a 1D NW structure using a mixture of N,N-dimethylformamide (DMF) and dimethyl sulfoxide (DMSO) in an atmospheric environment. The orientation sequence and optical properties of the NWs perovskite thin films can be effectively adjusted by changing the proportion of DMSO solvent. The obtained NWs exhibit the best order and dispersion, high light absorption, and good electrical properties. The PCE of the device based on the ordered NW perovskite film was 2.55%, which is 50% higher than that of the disordered NW perovskite film [109]. The photovoltaic performance of perovskite solar cells is affected by the morphology of the perovskite films. Cho et al. reported the preparation of two different morphologies of perovskite materials, compact (c-perovskite) and nanowire (NW-perovskite), using a simple two-step spin-coating process (Figure 14a). By spin coating a mixture of MAI, IPA, and DMF solution, PbI_2_ film was transformed into NW-perovskite after annealing at 100 °C. The NW perovskite films exhibited better light absorption, crystal structure, charge extraction, and photovoltaic performance than c-perovskite films. The PCE of the NW-perovskite device was as high as 16.8% [110]. Adding MAPbI_3_ NWs to a non-fullerene acceptor-based organic solar cell resulted in a PCE of 10.72%. The NWs incorporated into SnO_2_ and PBDB-T-SF:IT 4F could effectively resolve the incompatibility between the two dissimilar materials without affecting their properties and effectively improve the stability of solar cells [111]. As shown in Figure 14b, the (N-DPBI)-doped poly (P(NDI2OD-T2) polymer film as the electron transport layer (ETL) can minimize the resistance loss, enhance the electron extraction performance, and improve the surface coverage of the P(NDI2OD-T2) ETL on the MAPbI_3_ NW layer. The NW devices exhibited a PCE of up to 18.83% [112]. In addition to organic-inorganic perovskite NW solar cells, devices based on inorganic perovskite NW have also attracted attention from researchers. Kuang et al. synthesized inorganic CsPbI_3_ NWs on fluorine-doped tin oxide glass via a solution-dipping process and converted them into CsPbBr_3_ NWs via ion exchange. CsPbI_3_ and CsPbBr_3_ NWs were first used in perovskite solar cells, and their PCEs were 0.11% and 1.21%, respectively (Figure 14c,d). The inorganic CsPbBr_3_ NW solar cells exhibited good stability [113]. Perovskite NWs can not only be used as light-absorbing layers in solar cells. For example, CsPbBr_3_ NWs have been used to tune the surface electronic states of perovskite films to form composition-graded heterojunctions at perovskite/HTL interfaces. The obtained devices exhibited a high PCE of up to 21.4% (Figure 14e–g), and the incorporation of CsPbBr_3_ NWs enhanced the passivation of the bulk perovskite film surface and stabilized the PSCs [114]. 

## 4. Summary and Perspective

Halide perovskites are promising materials for future optoelectronic applications owing to their excellent optoelectronic properties. In particular, one-dimensional perovskite NWs exhibit few grain boundaries and low defect densities, making it possible to produce high-efficiency optoelectronic devices. This review summarizes the recent research progress into the controllable synthesis and application of halide perovskite NWs. The synthesis methods of perovskite NWs include HI, vapor-phase growth, and selfassembly, and their applications include lasers, PDs, and solar cells.

Despite significant progress in the synthesis and application of halide perovskite NWs, many problems still need to be solved to precisely control their size, morphology, and stability. First, the precise regulation of the synthesis size of perovskite NWs requires extensive research owing to the effects of reaction temperature, precursor ratio, and different ligands. The large aspect ratios and high uniformity of the perovskite NWs are essential for such devices. An increase in performance is critical. The morphologies of perovskite NWs obtained by different preparation methods are different, and the rough morphology and incomplete surface coverage limit their application in devices, thus reducing their performance.

The second problem is the instability of perovskite materials. Long-term exposure of perovskites to environmental factors such as water and oxygen significantly reduces their optoelectronic properties, thereby affecting their application in devices. This is also a critical issue for future development and application of perovskite materials.

Third, lead toxicity in lead-halide perovskites is an issue that must be addressed. Green and nontoxic perovskite materials substituted with other metal elements also have good development prospects.

Although significant progress has been made in the synthesis and application of halide perovskites, there is still room for improvement in device performance. For lasers and PDs, the performance of organic-inorganic perovskite NW devices is better than that of all-inorganic perovskite NW devices, but their stability is poor. Hence, high-performance all-inorganic perovskite NW devices still have great potential. In addition, the performance of NW-based solar cells is inferior to that of thin-film ones. There is an urgent need to develop excellent NW synthesis strategies for obtaining large-area and highly ordered NW arrays to improve the performance and environmental stability of solar cells. Therefore, there is still much research on the controllable synthesis and application of perovskite NWs to meet commercial needs.

## Figures and Tables

**Figure 1 nanomaterials-13-00419-f001:**
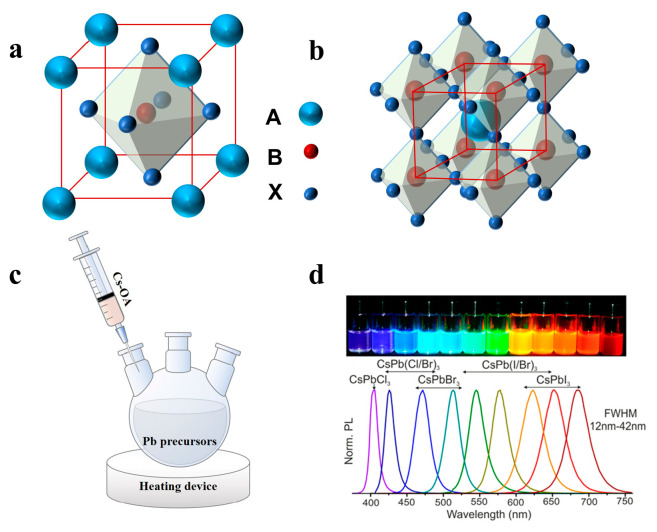
(**a**) One unit cell structure of ABX_3_ perovskites. (**b**) Extended network of ABX_3_ perovskites connected by corner-shared octahedrons. (**c**) Schematic illustrations of HI synthesis of colloidal CsPbX_3_ NCs. (**d**) Different colloidal dispersions in toluene under UV lamp (λ = 365 nm) and their representative PL spectra. Reprinted with permission from reference [37]. Copyright 2015, American Chemical Society.

**Figure 2 nanomaterials-13-00419-f002:**
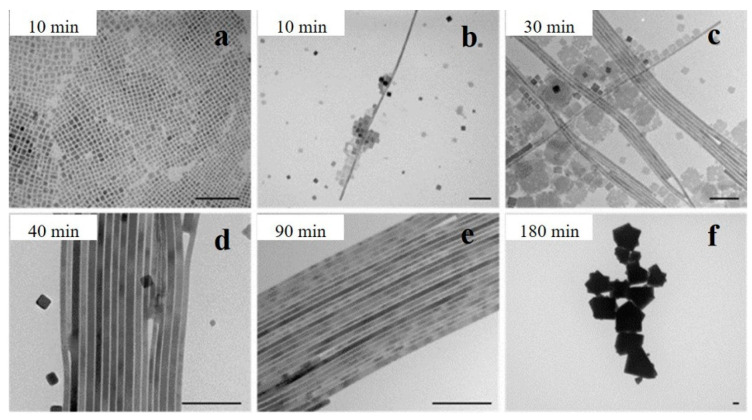
TEM images of the CsPbBr_3_ nanostructures synthesized with different reaction times: (**a**) less than 10 min, (**b**) after 10 min, (**c**) 30 min, (**d**) 40 min, (**e**) 90 min, and (**f**) 180 min. Scale bar: 100 nm. Reprinted with permission from reference [16]. Copyright 2015, American Chemical Society.

**Figure 3 nanomaterials-13-00419-f003:**
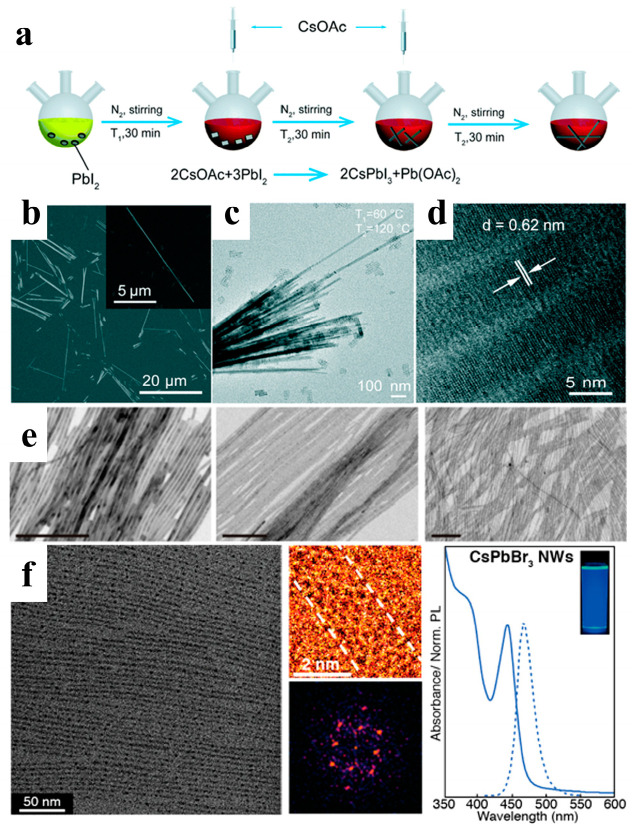
(**a**) Schematic diagram of the synthesis procedure with two injections at two corresponding temperatures: T1 = 60 °C and T2 = 120 °C. (**b**) Scanning electron microscope (SEM) image of the CsPbI_3_ NWs. The inset shows a high-magnification SEM image on a single CsPbI_3_ NW. (**c**) Transmission electron microscopy (TEM) image, and (**d**) high-resolution TEM image of the CsPbI_3_ NWs. Reprinted with permission from reference [38]. Copyright 2019, Royal Society of Chemistry. (**e**) TEM images of 10 nm, 5.1 nm, and 3.4 nm width NWs, respectively. Reprinted with permission from reference [39]. Copyright 2016, American Chemical Society. (**f**) Purified ultrathin CsPbBr_3_ NWs (scale bar: 50 nm); Diffraction pattern of a single CsPbBr_3_ NW; Absorption and PL spectra of CsPbBr_3_ NWs. Reprinted with permission from reference [40]. Copyright 2016, American Chemical Society.

**Figure 4 nanomaterials-13-00419-f004:**
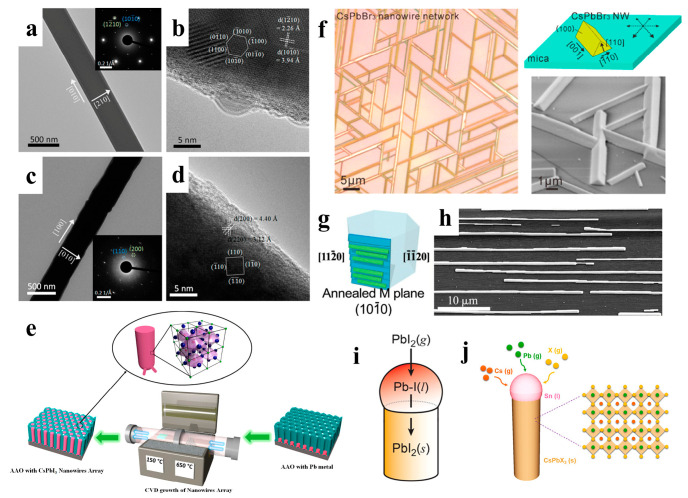
TEM and HRTEM images of (**a**,**b**) PbI_2_ NW and (**c**,**d**) MAPbI_3_ NW. Insets in (**a**,**c**) are their corresponding SAED patterns. Reproduced with permission from reference [41]. Copyright 2015, American Chemical Society. (**e**) Illustration of a vapor-phase method for growing perovskite NWs in anodic aluminum membranes. Reproduced with permission from reference [42]. Copyright 2017, American Chemical Society. (**f**) Illustration of the vapor-phase epitaxial growth of CsPbBr_3_ NWs with a triangular cross-section. Reproduced with permission from reference [43]. Copyright 2016, American Chemical Society. (**g**) Graphoepitaxial CsPbBr_3_ NWs and SEM images of (**h**) an assembly of NWs. Reproduced with permission from references [48,53]. Copyright 2017, American Chemical Society. (**i**) Synthesis of PbI_2_ NWs. Reproduced with permission from reference [45] Copyright 2017, American Chemical Society. (**j**) Schematic diagram illustrating the VLS growth process of CsPbX_3_ NW using the Sn catalysts. Reproduced with permission from reference [47]. Copyright 2019, American Chemical Society.

**Figure 5 nanomaterials-13-00419-f005:**
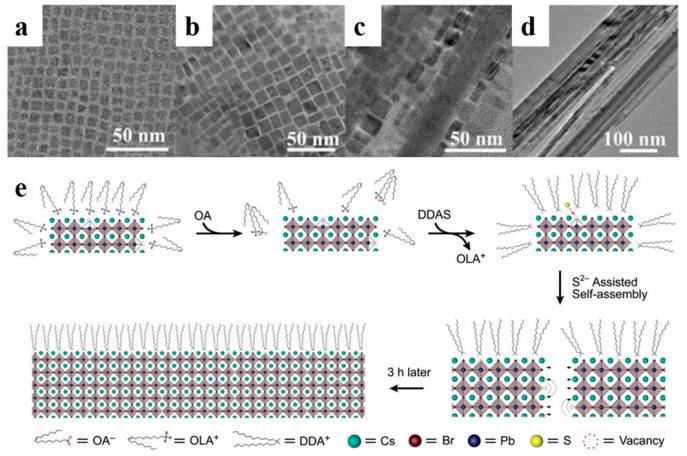
TEM images of samples obtained with different reaction times: (**a**) pristine HVPQDs, (**b**) 5 min, (**c**) 60 min, and (**d**) 180 min. (**e**) QD selfassembly mechanism. Reproduced with permission from reference [49]. Copyright 2019, John Wiley and Sons.

**Figure 6 nanomaterials-13-00419-f006:**
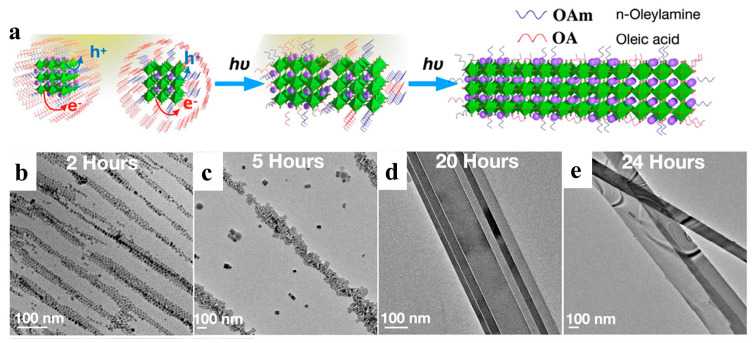
(**a**) Schematic illustration of the light-induced selfassembly growth process. (**b**–**e**) Shape evolution of 1D CsPbBr_3_ NWs with different illumination times under a light intensity of 1.7 suns: (**b**) NCs formed after 2 h, (**c**) nanorods formed after 5 h, (**d**) NWs formed after 20 h, and (**e**) NWs formed after 24 h. Reproduced with permission from reference [50]. Copyright 2019, American Chemical Society.

**Figure 7 nanomaterials-13-00419-f007:**
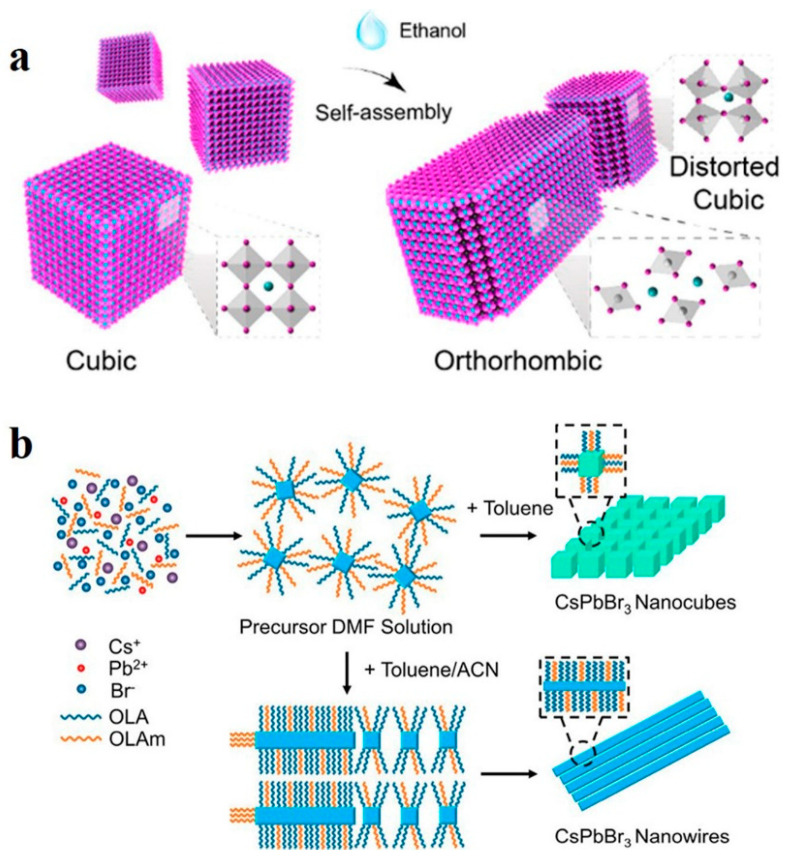
(**a**) Selfassembly mechanism of CsPbI_3_ nanocubes into NWs. Reproduced with permission from reference [51]. Copyright 2018, American Chemical Society. (**b**) Formation mechanism of CsPbBr_3_ NWs in toluene/ACN. Reproduced with permission from reference [52]. Copyright 2019, American Chemical Society.

**Figure 8 nanomaterials-13-00419-f008:**
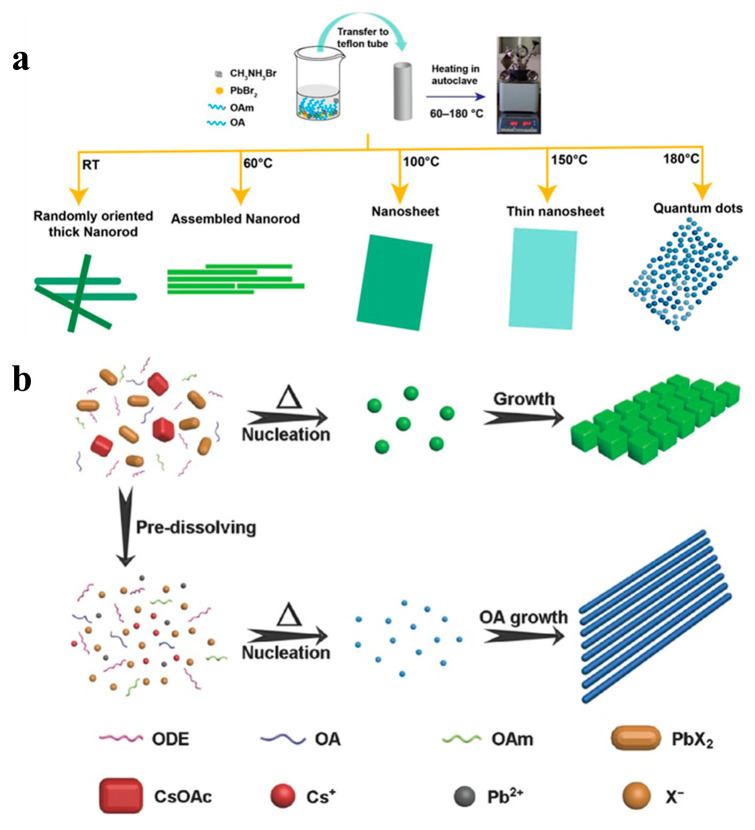
(**a**) Schematic diagram of the synthesis procedure of MAPbBr_3_ nanostructures and their structural evolution with different reaction temperatures. Reproduced with permission from reference [85]. Copyright 2020, American Chemical Society. (**b**) Proposed growth process of CsPbBr_3_ NCs obtained without and with precursor dissolving. Reproduced with permission from reference [53]. Copyright 2017, John Wiley and Sons.

**Figure 9 nanomaterials-13-00419-f009:**
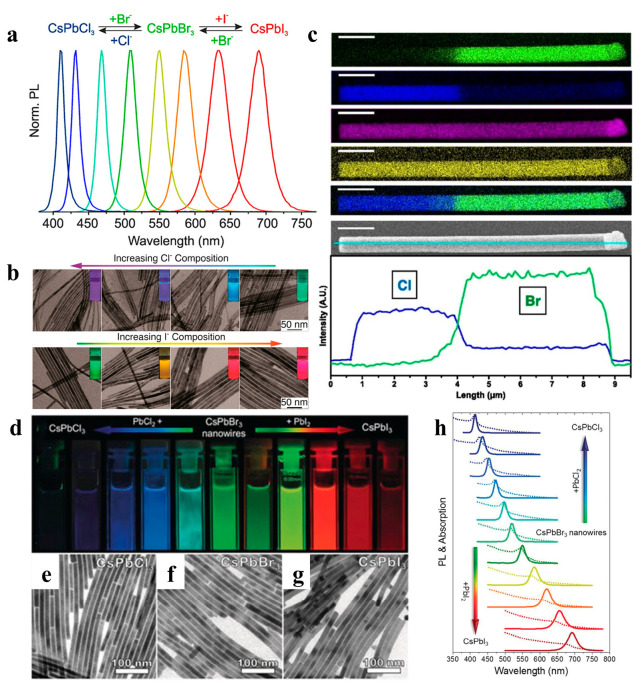
(**a**) PL spectra of the anion exchange reaction in the CsPbX_3_ NCs system. Reproduced with permission from reference [86]. Copyright 2015, American Chemical Society. (**b**) TEM images of CsPbX_3_ NWs with various conversion degrees. The insets are the CsPb(Br/Cl)_3_ and CsPb(Br/I)_3_ NWs illuminated under 365 nm. Reproduced with permission from reference [28]. Copyright 2016, American Chemical Society. (**c**) EDS mapping and SEM image of a single heterojunction NW. Reproduced with permission from reference [87]. Copyright 2017, PNAS. (**d**) Photograph of colloidal dispersions of CsPbX_3_ perovskite NWs under UV irradiation (λ = 367 nm). (**e**–**g**) Bright-field TEM images of CsPbCl_3_, CsPbBr_3_, and CsPbI_3_ NWs. (**h**) Corresponding UV/Vis absorption (dotted lines) and PL (solid lines) spectra of colloidal CsPbX_3_ perovskite NWs. Reproduced with permission from reference [54]. Copyright 2017, John Wiley and Sons.

**Figure 10 nanomaterials-13-00419-f010:**
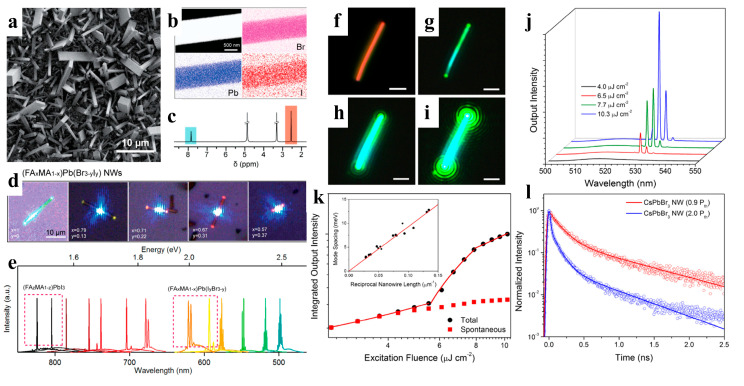
(**a**) SEM image of (FA_0_._67_MA_0_._33_)Pb(Br_2_._69_I_0_._31_) NWs. (**b**) EDS mapping of a single NW. (**c**) 1H NMR spectrum showing the MA and FA in the NWs. (**d**) Optical images of (FA_x_MA_1−x_)Pb(Br_3−y_I_y_) NWs excited by a laser with a 442 nm wavelength. (**e**) Broad wavelength-tunable lasing measured from single-crystal NWs. Reproduced with permission from reference [93]. Copyright 2016, American Chemical Society. (**f**) Dark-field image of a CsPbBr_3_ NW. (**g**–**i**) The NW of (**f**) under excitation from a femtosecond pulsed laser with increasing excitation fluence. All scale bars are 2 μm. (**j**) Power-dependent emission measured for the CsPbBr_3_ NW of (**f**–**i**). The narrow emission peak at approximately 530 nm indicates a lasing characteristic of the CsPbBr_3_ NW. (**k**) The integrated output intensity of the CsPbBr_3_ NW under different pump fluences. The inset is a plot of inverse NW length versus mode spacing. (**l**) TRPL of a CsPbBr_3_ NW under different lasing thresholds. Reproduced with permission from reference [8]. Copyright 2016, PNAS.

**Figure 11 nanomaterials-13-00419-f011:**
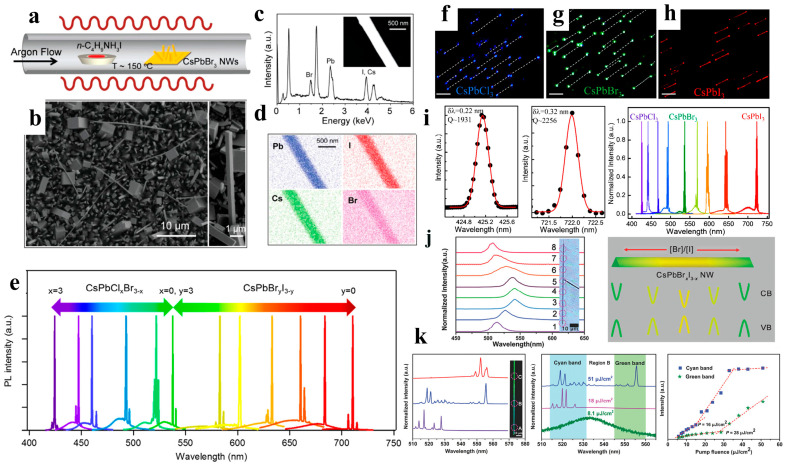
(**a**) Schematic illustration of the vapor conversion of CsPbBr_3_ into CsPb(Br,I)_3_ by introducing n-butylammonium iodide at approximately 150 °C. (**b**) SEM image of the obtained CsPb(Br,I)_3_, reacted for 2 min. The right inset shows a single NW with a rectangular flat end-facet. (**c**) EDX measurement result of an individual NW (inset). (**d**) EDX mapping of the corresponding NW. (**e**) Tunable lasing characteristic of CsPbX_3_ single-crystal NWs. Reproduced with permission from reference [17]. Copyright 2016, American Chemical Society. (**f**–**h**) Optical images of CsPbCl_3_, CsPbBr_3_, and CsPbI_3_ NW in the in-plane direction (scale bar: 10 μm). (**i**) Gaussian fitting of the lasing spectra detected from CsPbCl_3_ (left) and CsPbI_3_ (middle) NWs. The broad wavelength tunable lasing spectra of CsPbX_3_ and alloy NWs (right). Reproduced with permission from reference [95]. Copyright 2016, American Chemical Society. (**j**) Normalized µ-PL spectra (left) and bandgap schematic diagram (right) of the composition-graded CsPbBr_x_I_3−x_ NW. (**k**) µ-PL spectra of the CsPbBr_x_I_3−x_ NW at different positions measured above the threshold. Reproduced with permission from reference [96]. Copyright 2016, American Chemical Society.

**Figure 12 nanomaterials-13-00419-f012:**
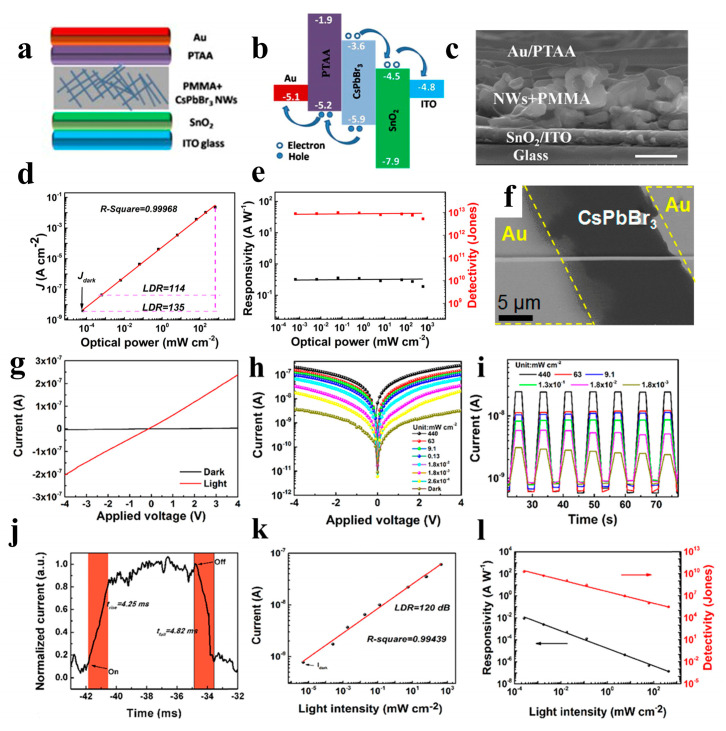
(**a**) Device structure of the perovskite NW PD. (**b**) Relative band-edges of the films in the device. (**c**) Cross-sectional SEM image of the device based on perovskite NW PD (scale bar: 500 nm). (**d**) Linear dynamic range of the PD. (**e**) The responsivity and detectivity of the perovskite PD. Reproduced with permission from reference [98]. Copyright 2018, Elsevier. (**f**) SEM image of the as-fabricated device based on a single CsPbBr_3_ NW. Reproduced with permission from reference [99]. Copyright 2016, American Chemical Society. (**g**) I − V curves of the CsPb_0_._84_Sn_0_._16_(I_0_._008_Br_0_._992_)_3_ perovskite NWs. (**h**) I − V and (**i**) I − t curves of the perovskite NWs under different light intensities. (**j**) Time response of the PDs based on perovskite NWs. (**k**) The current versus light intensity of the PD. (**l**) Responsivity and detectivity of the device. Reproduced with permission from reference [100]. Copyright 2020, American Chemical Society.

**Figure 13 nanomaterials-13-00419-f013:**
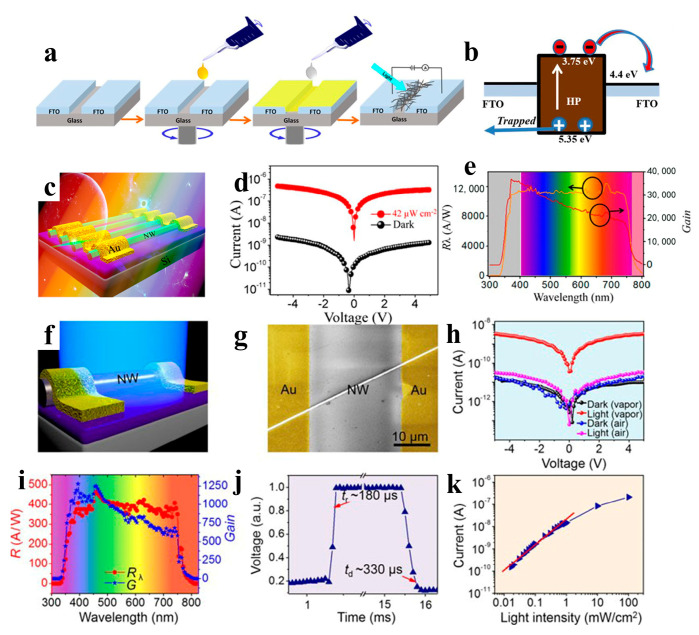
(**a**) Two-step all-solution synthesis of NW-based PD. (**b**) Energy band alignment of the HP and FTO films in a device under illumination. Reproduced with permission from reference [102]. Copyright 2018, Elsevier. (**c**) Schematic diagram of PD fabricated with MAPbI_3_ perovskite NW array. (**d**) I − V measurement result of the PD under the conditions of darkness and illumination. The wavelength of light is 550 nm with 42 μW cm^−2^ intensity. (**e**) The responsivity and photoconductive gain of PD versus the wavelength of light. Reproduced with permission from reference [103]. Copyright 2017, American Chemical Society. (**f**) Schematic diagram of the PD based on a single MAPbI_3_ NW. (**g**) SEM image of the PD. (**h**) Dark/photocurrent of the PD fabricated with MAPbI_3_ NW prepared under vapor and air conditions (light intensity is 60 μW cm^−2^). (**i**) The responsivity and photoconductive gain of the PD versus light wavelength. (**j**) The rise or decay edges of the PD. (**k**) Photocurrent of PD under different light intensities. Reproduced with permission from reference [104]. Copyright 2018, American Chemical Society.

**Figure 14 nanomaterials-13-00419-f014:**
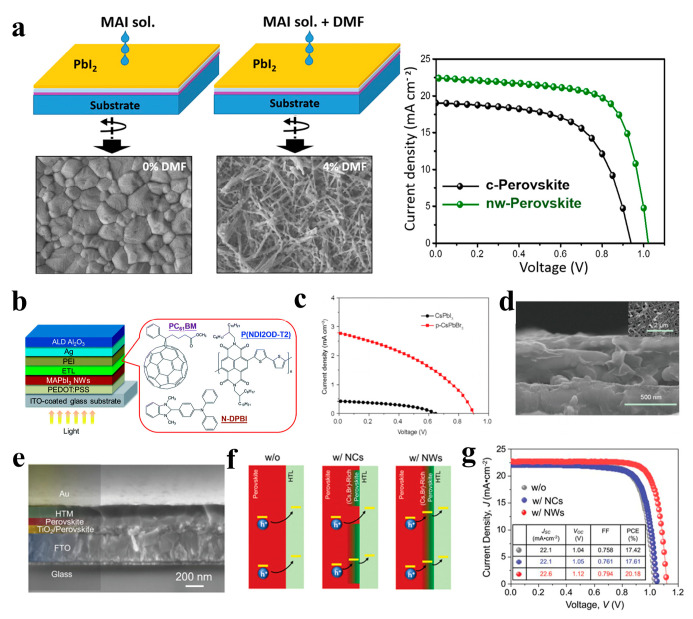
(**a**) Schematic diagram of the two-step preparation and corresponding SEM image of the compact perovskite (c-perovskite) and NW-perovskite (left). J − V curves of the PSC devices (right). Reproduced with permission from reference [110]. Copyright 2018, Elsevier. (**b**) Schematic illustration of the device architecture and the molecular structures of ETL materials. Reproduced with permission from reference [112]. Copyright 2017, Royal Society of Chemistry. (**c**) J − V characteristics and (**d**) SEM images of the PSCs based on CsPbBr_3_ NWs. Reproduced with permission from reference [113]. Copyright 2017, Springer Nature. (**e**) Cross-sectional SEM image of the device based on CsPbBr_3_ NW-incorporated MAPbI_3_ thin film. (**f**) Schematic illustration of the hole-extraction in the devices without CsPbBr_3_ NCs (left), with CsPbBr_3_ NCs (middle), and with CsPbBr_3_ NWs (right) incorporation. (**g**) J − V curves of the devices. Reproduced with permission from reference [114]. Copyright 2019, John Wiley and Sons.

**Table 1 nanomaterials-13-00419-t001:** Typical NW parameters obtained from different synthetic methods.

Perovskite NWs	Synthesis	PL Range (nm)	Diameter (nm)	Length (μm)	Ref.
CsPbBr_3_	HI	-	~12	5	[16]
CsPbI_3_	HI	~685	5–80	10–20	[38]
CsPbBr_3_	HI	473, 483	3.4 ± 0.5	-	[39]
CsPbBr_3_	HI	465	2.2 ± 0.2	Several microns	[40]
MAPbI_3_	Vapor	-	~200	~10	[41]
CsPbI_3_	Vapor	-	200–250	~1	[42]
CsPbBr_3_	Vapor	528	~1 μm	Tens of micrometers	[43]
CsPbBr_3_	Vapor	~520	20–2000	-	[44]
MAPbI_3_	VLS	768	100–2000	Tens of micrometers	[45]
MAPbI_3_	VLS	-	-	-	[46]
CsPbX_3_	VLS	422, 539, 740	~150	-	[47]
MAPbBr_3_	Ligand	532	~100	2–4.5	[48]
CsPbBr_3_	Ligand	525	20–60	Several millimeters	[49]
CsPbBr_3_	Light	-	Hundreds of nanometers	10–20	[50]
CsPbI_3_	Polar solvent	690	Up to submicron	Several micrometers	[51]
CsPbBr_3_	Polar solvent	-	-	-	[52]
CsPbX_3_	Solvothermal	410–700	~2.6	-	[53]
CsPbX_3_	Anion Exchange	409–680	10 ± 2	-	[28]
CsPbX_3_	Anion Exchange	400–700	~12	-	[54]

**Table 2 nanomaterials-13-00419-t002:** The parameters of different devices based on halide perovskite NWs.

Material	Devices	Synthesis Method	Wavelength [nm]		Threshold [μJ cm^−2^]	Quality Factor	Ref.
MAPbX_3_ NWs	Lasers	surface-initiated solution growth	~490–~770		0.22	3600	[91]
MAPbX_3_ NWs	Lasers	one-step in situ solution	548		375	589	[92]
FAPbX_3_ NWs	Lasers	low-temperature solution growth	490–824		~Several	~2000	[93]
CsPbX_3_ NWs	Lasers	low-temperature solution growth	~430, ~532		5	1009 ± 5	[8]
CsPbX_3_ NWs	Lasers	solution growth	~425–~722		4	2256	[17]
CsPbCl_3−3x_Br_3x_ NWs	Lasers	chemical vapor deposition	480–525		11.7–35	-	[94]
CsPbX_3_ NWs	Lasers	chemical vapor deposition	425, 525, 725		6	1300	[95]
CsPbBr_x_I_3−x_ NWs	Lasers	vapor-phase epitaxial growth	521, 556		16, 28	-	[96]
CsPbBr_3_ NWs	Lasers	wet chemical approach	~526		13	7860	[97]
Material	Devices	Synthesis Method	Responsivity [A W^−1^]		Rise Time	Decay Time	Ref.
CsPbBr_3_ NWs	PDs	solution-phase process and halide exchange	0.3		0.4 ms	0.43 ms	[98]
CsPbBr_3_ NWs	PDs	graphoepitaxial growth	4400		252 μs	300 μs	[99]
CsPb_x_Sn_1−x_(Br_y_I_1−y_)_3_ NWs	PDs	two-step solution method	8.5 × 10^−3^		4.25 ms	4.82 ms	[100]
CsSnI_3_ NWs	PDs	hot-pressure welding	9.9 × 10^−3^		446 ms	534 ms	[101]
MAPbX_3_ NWs	PDs	two-step solution method	0.23		53.2 μs	50.2 μs	[102]
MAPb(I_1–x_Br_x_)_3_ NWs	PDs	a fluid-guided antisolvent vapor-assisted crystallization	12,500		0.34 μs	0.42 μs	[103]
MAPbI_3_ NWs	PDs	saturated vapor-assisted crystallization	460		180 μs	330 μs	[104]
MAPbI_3_ NWs	PDs	one step selfassembly	1.32		0.2 μs	0.3 μs	[105]
(MTEA)_2_(MA)_n−1_PbnI_3n+1_ NWs	PDs	capillary-bridge lithography	7.3 × 10^3^		40 μs	52.2 μs	[106]
MAPbI_3_ NWs	PDs	two-step spin-coating	37.14		91 μs	563 μs	[107]
Material	Devices	Synthesis Method	J_SC_ (mA cm^−2^)	V_OC_ (V)	FF	PCE (%)	Ref.
MAPbI_3_ NWs	PSCs	two-step spin-coating	19.12	1.052	0.712	14.71	[108]
MAPbI_3_ NWs	PSCs	two-step spin-coating	9.1	0.98	28.65	2.55	[109]
MAPbI_3_ NWs	PSCs	two-step spin-coating	22.6	1.03	71.6	16.8	[110]
MAPbI_3_ NWs	PSCs	two-step sequential deposition	23.92	0.819	0.547	10.72	[111]
MAPbI_3_ NWs	PSCs	two-step spin-coating	21.39	1.01	0.80	18.83	[112]
CsPbI_3_ NWs	PSCs	solution-dipping process	0.415	0.643	0.429	0.11	[113]
CsPbBr_3_ NWs	PSCs	solution-dipping process	2.96	0.851	0.445	1.21	[113]
CsPbBr_3_ NWs	PSCs	solution growth	24.1	1.12	0.791	21.4	[114]

## Data Availability

Data sharing is not applicable to this article as no datasets were generated or analyzed during the current study.

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
