# Peer review of "Metal Halide Perovskite Nanowires: Controllable Synthesis, Mechanism, and Application in Optoelectronic Devices"

_nanomaterials, 2023, doi:10.3390/nano13030419_

Round 1

Reviewer 1 Report

Reviewers' comments:

In this paper, the authors reported a summary of synthetic strategies and mechanisms of one-dimensional (1D) halide perovskite nanowires (NWs) and also give a short perspective applications 1D PVSK NWs for optoelectronics, including lasers, photodetectors, and solar cells.

Comments

1.     For readability, please improve the resolution of all figures.

2.     The author summarized several methods in section 2. Could the authors give a short comment for mass product ability for the usage of device applications in this section? Which methods have the most potential for commercialization?

3.     For readability, please keep the integrity of the Table. Do not let the table separate into 2 pages. For example: Table 2.

4.     Comparing to the conventional perovskite devices (eg. planar structure), the performance of NWs-type device is lower than conventional ones. For example, the conventional solar cell, Lee and his group use the thermal-assisted blade-coating method, which can be used for large area production, to obtain >21% efficiency cell. (https://doi.org/10.1016/j.cej.2021.131609) The blade-coated perovskite films are fabricated on a flexible substrate to obtain >300 cm2 and >20% efficiency solar module. (https://doi.org/10.1002/smtd.202200669 ) on the other hand, there are many studies via large-area fabricability techs, such as slot-die coating (https://doi.org/10.1002/admi.202100743), flexographic printing (DOI10.1002/admt.202101282), and screen printing (Nature Communications volume 8, Article number: 15684 (2017)), to approach high efficiency and a large area for commercialized application. Could the authors give a short comment to explain this issue?

5.     Would the authors please add one column of the NWs synthesis technology to Table 2? And strengthen the interconnectedness between synthesis technology and applications in the context.

6.     For NWs laser, the behavior of photonic and hybrid plasmonic modes (https://doi.org/10.1002/adom.202200603) is an interesting topic to be discussed. The author may describe some of this behavior in the manuscript.

7.     The authors should list/reference more studies of laser, PDs and photovoltaics applications in Table 2. I think the refence you cited is a lite bit old. There is a huge improvement in recent years (such as in 2021 and 2022). Please summarized them in this review paper.  For example, Wu achieve an ultra-high average responsivity of 7.3 × 103 A W1 of strongly interacted layered-perovskite nanowires photodetectors. (https://doi.org/10.1021/acsami.1c20851)

Author Response

Response to Reviewer

In this paper, the authors reported a summary of synthetic strategies and mechanisms of one-dimensional (1D) halide perovskite nanowires (NWs) and also give a short perspective applications 1D PVSK NWs for optoelectronics, including lasers, photodetectors, and solar cells.

Response: We thank the reviewer for the detailed suggestions to improve the quality of our manuscript. We have addressed the issues point-to-point in the revised manuscript. Our responses to the reviewer’s comments are listed as followings.

  1. For readability, please improve the resolution of all figures.

Response: Thanks for the suggestion. We have improved the resolution of all figures.

  1. The author summarized several methods in section 2. Could the authors give a short comment for mass product ability for the usage of device applications in this section? Which methods have the most potential for commercialization?

Response: Thanks for your suggestion. We have summarized the performance of different synthesis methods for device applications at the beginning of chapter 2.

Most perovskite NWs are synthesized by solution processes, such as hot injection, self-assembly, solvothermal, and anion exchange, exhibiting the advantages of high yield, aspect ratio regulation, large area, and easy transfer to the device. Vapor-phase growth provides high-quality crystals and low defect density, thus effectively improving the performance of optoelectronic devices.[36] In this section, we review the synthetic strategies of perovskite NWs in recent years, including the hot injection method, vapor phase synthesis method, and self-assembly induced by different factors. The morphological differences between NWs synthesized using each method are summarized in Table 1.

[36] Zhang, D.; Zhang, Q.; Zhu, Y.; Poddar, S.; Zhang, Y.; Gu, L.; Zeng, H.; Fan, Z. Metal halide perovskite nanowires: synthesis, integration, properties, and applications in optoelectronics. Adv. Energy Mater. 2022, 2201735.

  1. For readability, please keep the integrity of the Table. Do not let the table separate into 2 pages. For example: Table 2.

Response: Thanks for indicating our mistake. Table 2 has been updated to display on a full page.

  1. Comparing to the conventional perovskite devices (eg. planar structure), the performance of NWs-type device is lower than conventional ones. For example, the conventional solar cell, Lee and his group use the thermal-assisted blade-coating method, which can be used for large area production, to obtain > 21% efficiency cell. (https://doi.org/10.1016/j.cej.2021.131609) The blade-coated perovskite films are fabricated on a flexible substrate to obtain > 300 cm2 and > 20% efficiency solar module. (https://doi.org/10.1002/smtd.202200669) on the other hand, there are many studies via large-area fabricability techs, such as slot-die coating (https://doi.org/10.1002/admi.202100743), flexographic printing (DOI10.1002/admt.202101282), and screen printing (Nature Communications volume 8, Article number: 15684 (2017)), to approach high efficiency and a large area for commercialized application. Could the authors give a short comment to explain this issue?

Response: Thanks for the valuable suggestion. In theory, 1) perovskite NWs show larger specific surface area than conventional polycrystalline perovskite films, which allows them to capture more solar energy; 2) perovskite NWs with 1D structure have few ionic defects and grain boundaries, which contributes to better carrier extraction and transport. (ACS Appl. Energy Mater. 2022, 5, 1342−1377.) However, in practice, the performance of PSCs based on NWs is usually inferior to that of polycrystalline perovskite devices, which can be ascribed to the following facts. 1) Almost all NW-based PSCs exhibit a random orientation between electron transport layers and hole transport layers, thus losing the advantages mentioned above. 2) The researches on NW-type device are very lacking, and further improvements in NW-based PSCs are rarely reported, while the performance of conventional PSCs received booming development, including defects passivation, energy level modulation, and crystalline engineering.

  1. Would the authors please add one column of the NWs synthesis technology to Table 2? And strengthen the interconnectedness between synthesis technology and applications in the context.

Response: Thanks for the suggestion. We have added one column in Table 2 to strengthen the interconnectedness between synthesis technology and applications.

  1. For NWs laser, the behavior of photonic and hybrid plasmonic modes (https://doi.org/10.1002/adom.202200603) is an interesting topic to be discussed. The author may describe some of this behavior in the manuscript.

Response: Thanks for the suggestion. We have added the behavior of photonic and hybrid plasmonic modes in the revised manuscript.

Various mechanisms have been proposed to explain the perovskite lasers, such as the plasma mode and photon mode.[15,91,95-97] Plasma lasers typically have lower thresholds than those of conventional photon lasers.[95,98] Jin et al. introduced a single crystal MAPbI3 NWs into a plasma laser as an organic-inorganic semiconductor gain material, achieving a lower threshold (13.5μJ cm−2) under ambient conditions.[98] Lu et al. recently presented a 4f measurement system based on the reconstruction of the near-field of hybrid plasmonic MAPbBr3 perovskite nanolasers to determine the exact resonant mode.[95] The scaling laws for wire-type hybrid plasmonic perovskite lasers were successfully determined by combining them with other optoelectronic measurements. For example, when a perovskite wire with a width less than 2.5 µm, a hybrid plasmonic perovskite laser displayed low power consumption and strong light-matter interaction with a maximum group index of 24.2, superior to that of a photonic laser.

[15] Wang, Y.; Li, X.; Song, J.; Xiao, L.; Zeng, H.; Sun, H. Allinorganic colloidal perovskite quantum dots: a new class of lasing materials with favorable characteristics. Adv. Mater. 2015, 27, 7101-7108.

[91] Zhu, H.; Fu, Y.; Meng, F.; Wu, X.; Gong, Z.; Ding, Q.; Gustafsson, M.V.; Trinh, M.T.; Jin, S.; Zhu, X. Lead halide perovskite nanowire lasers with low lasing thresholds and high quality factors. Nat. Mater. 2015, 14, 636-642.

[95] Huang, Z.T.; Chen, J.W.; Li, H.; Zhu, Y.; Cui, Q.; Xu, C.; Lu, T.C. Scaling laws for perovskite nanolasers with photonic and hybrid plasmonic modes. Adv. Opt. Mater. 2022, 10, 2200603.

[96] Wang, J.; Jia, X.; Guan, Y.; Ren, K.; Yu, H.; Wang, Z.; Qu, S.; Yang, Q.; Lin, J.; Wang, Z. The electron–hole plasma contributes to both plasmonic and photonic lasing from CH3NH3PbBr3 nanowires at room temperature. Laser Photonics Reviews 2021, 15, 2000512.

[97] Shang, Q.; Zhang, S.; Liu, Z.; Chen, J.; Yang, P.; Li, C.; Li, W.; Zhang, Y.; Xiong, Q.; Liu, X. Surface plasmon enhanced strong exciton–photon coupling in hybrid inorganic–organic perovskite nanowires. Nano Lett. 2018, 18, 3335-3343.

[98] Yu, H.; Ren, K.; Wu, Q.; Wang, J.; Lin, J.; Wang, Z.; Xu, J.; Oulton, R.F.; Qu, S.; Jin, P. Organic–inorganic perovskite plasmonic nanowire lasers with a low threshold and a good thermal stability. Nanoscale 2016, 8, 19536-19540.

  1. The authors should list/reference more studies of laser, PDs and photovoltaics applications in Table 2. I think the refence you cited is a lite bit old. There is a huge improvement in recent years (such as in 2021 and 2022). Please summarized them in this review paper. For example, Wu achieve an ultra-high average responsivity of 7.3 × 103 A W–1 of strongly interacted layered-perovskite nanowires photodetectors. (https://doi.org/10.1021/acsami.1c20851)

Response: Thanks for the suggestion. We have added some recent works about lasers, PDs, and SCs in Chapter 3.

Lasers:

Qu et al. proposed a metal-cavity plasma nanolaser based on a single Au-MAPbBr3 NW prepared using a one-step in-situ solution. By changing the size of the hybrid plasma nanolaser (150 - 500 nm), the competition between the photonic and plasmonic modes was observed. The nanolasers exhibited attractive lasing behavior with a threshold of 375 μJ cm-2 and Q value of 589.[92]

CsPbBr3 NWs were integrated with nanostructured indium tin oxide substrates, which exhibit near-unity refractive indices and high electrical conductivities. This approach effectively reduced the laser mode leakage, and the NWs device exhibited a high Q factor of 7860 at a low optical pumping threshold (13 μJ cm-2). [102]

[92] Jia, X.; Wang, J.; Huang, Z.; Chu, K.; Ren, K.; Sun, M.; Wang, Z.; Jin, P.; Liu, K.; Qu, S. Metallic cavity nanolasers at the visible wavelength based on in situ solution-grown Au-coated perovskite nanowires. J. Mater. Chem. C 2022, 10, 680-687.

[102] Markina, D.I.; Pushkarev, A.P.; Shishkin, I.I.; Komissarenko, F.E.; Berestennikov, A.S.; Pavluchenko, A.S.; Smirnova, I.P.; Markov, L.K.; Vengris, M.; Zakhidov, A.A. Perovskite nanowire lasers on low-refractive-index conductive substrate for high-Q and low-threshold operation. Nanophotonics 2020, 9, 3977-3984.

PDs:

Zhou et al. adopted the hot-pressing welding method to reduce the interface and improve the quality of the CsSnI3 NWs. A high-performance NW PD was obtained with a responsivity of 9.9 × 10-3 A W-1 and a detectivity of 7.2 × 108 Jones. [110]

Wu et al. reported layered perovskite (MTEA)2(MA)n-1PbnI3n+1 single-crystal NW arrays. The sulfur-sulfur interaction between alkylammonium 2-(methylthio) ethylamine (MTEA) cations promotes crystallization by controlling the nucleation and growth of perovskites in capillary bridges, and NW arrays with single crystallinity and pure crystallographic orientation were obtained. When the crystallinity of layered perovskites (n = 3) is enhanced, ultrasensitive PDs based on NW arrays exhibit a responsivity of 7.3×103 A W-1 and a specific detectivity of 3.9×1015 Jones.[116] Adding 1-butyl-3-methylimidazolium tetrafluoroborate (BMIMBF4) as an additive to MAPbI3 NWs improved device performance owing to fewer defects, better crystallization, enhanced light absorption, and improved charge transfer in MAPbI3 NWs. The NW PD exhibited remarkable performance with detectivity, linear detection range, and noise-equivalent power of 2.06 × 1013 Jones, 160 dB, and 1.38 × 10-15 W Hz-1/2, respectively, and the stability of the device was effectively improved. [117]

[110] Zhou, H.; Tang, X.; Gao, Z. Lead-less perovskite alloy nanowire photodetector with high performance. Colloid Interface Science Communications 2022, 49, 100638.

[116] Yuan, M.; Zhao, Y.; Feng, J.; Gao, H.; Zhao, J.; Jiang, L.; Wu, Y. Ultrasensitive photodetectors based on strongly interacted layered-perovskite nanowires. ACS Appl. Mater. Interfaces 2022, 14, 1601-1608.

[117] Wu, D.; Xu, Y.; Zhou, H.; Feng, X.; Zhang, J.; Pan, X.; Gao, Z.; Wang, R.; Ma, G.; Tao, L. Ultrasensitive, flexible perovskite nanowire photodetectors with longterm stability exceeding 5000 h. InfoMat 2022, e12320.

PSCs:

Adding MAPbI3 NWs to a non-fullerene acceptor-based organic solar cell resulted in a PCE of 10.72%. The NWs incorporated into SnO2 and PBDB-T-SF:IT 4F could effectively resolve the incompatibility between the two dissimilar materials without affecting their properties and effectively improve the stability of solar cells.[124] As shown in Figure 14b, the (N-DPBI)-doped poly (P(NDI2OD-T2) polymer film as the electron transport layer (ETL) can minimize the resistance loss, enhance the electron extraction performance, and improve the surface coverage of the P(NDI2OD-T2) ETL on the MAPbI3 NW layer. The NW devices exhibited a PCE of up to 18.83%.[125]

[124] Zhao, F.; Deng, L.; Wang, K.; Han, C.; Liu, Z.; Yu, H.; Li, J.; Hu, B. Surface modification of SnO2 via MAPbI3 nanowires for a highly efficient non-fullerene acceptor-based organic solar cell. ACS applied materials interfaces 2020, 12, 5120-5127.

[125] Chang, C.-Y.; Tsai, B.-C.; Lin, M.-Z.; Huang, Y.-C.; Tsao, C.-S. An integrated approach towards the fabrication of highly efficient and long-term stable perovskite nanowire solar cells. J. Mater. Chem. A 2017, 5, 22824-22833.

Reviewer 2 Report

This manuscript reviews the synthetic strategies and mechanisms of halide perovskite NWs, including hot injection, vapor phase growth, self-assembly, and solvothermal. They also summarize their attractive applications in optoelectronics, including lasers, photodetectors, and solar cells. The topic is timely and would be attractive to a broad range of readers.

1)      In the second paragraph of the introduction, instead of referring a paper, the authors can consider explaining the primary advantages of NWs and why they are attractive for optoelectronic devices.

2)      For the ‘Hot injection’ and ‘Vapor growth’ parts, it would be better to provide a simple introduction of the basic theory and procedure in the beginning.

3)      The author mention that ‘Considering the optoelectronic properties, NWs grew in the vapor phase generally have higher quality and lower defect density’. Is this compared to the ‘hot injection’ method? More detailed explanations are needed.

4)      The authors can discuss current challenges in the application of perovskite NWs on lasers, PDs, and solar cells at the end of each part or in the Summary and Perspective.

5)      The author mention that ‘the low-dimensional MAPbI3 is expected to outperform the three-dimensional structure, improving the hole migration from perovskite to hole transport layer (HTL) and separation at HTL/perovskite interface’. Would this be applicable for QDs or just limited to NWs?

Author Response

Response to Reviewer

This manuscript reviews the synthetic strategies and mechanisms of halide perovskite NWs, including hot injection, vapor phase growth, self-assembly, and solvothermal. They also summarize their attractive applications in optoelectronics, including lasers, photodetectors, and solar cells. The topic is timely and would be attractive to a broad range of readers.

Response: We thank the reviewer for the detailed suggestions. We have addressed the issues point-to-point in the revised manuscript. Our responses to the reviewer’s comments are listed as followings.

  1. In the second paragraph of the introduction, instead of referring a paper, the authors can consider explaining the primary advantages of NWs and why they are attractive for optoelectronic devices.

Response: We thank the reviewer for the comments. We supplement and explain the main advantages of NWs and why they are attractive for optoelectronic devices. The related content is in the second paragraph of the first chapter of the revised manuscript.

Compared with their thin films or bulk counterparts, perovskite NWs have many advantages, including negligible ionic defects and grain boundaries, and thus exhibit enhanced photogenerated carrier transport properties, which can significantly improve the performance, reliability, and stability of optoelectronic devices.[18,20] Furthermore, the inherently large surface area of perovskite NWs significantly enhances their light-harvesting properties. The spatial confinement of charge carriers in the highly crystalline 1D structure further improves charge separation, extraction, and transport.[30] These advantages make 1D perovskite NWs promising materials for next-generation photovoltaic applications.

[18]Kumar, G.S.; Sumukam, R.R.; Rajaboina, R.K.; Savu, R.N.; Srinivas, M.; Banavoth, M. Perovskite nanowires for next-generation optoelectronic devices: Lab to Fab. ACS Appl. Energy Mater. 2022, 5, 1342-1377.

[20]Hong, K.; Le, Q.V.; Kim, S.Y.; Jang, H.W. Low-dimensional halide perovskites: review and issues. J. Mater. Chem. C 2018, 6, 2189-2209.

[30]Quan, L.N.; Kang, J.; Ning, C.-Z.; Yang, P.J.C.r. Nanowires for photonics. 2019, 119, 9153-9169.

  1. For the ‘Hot injection’ and ‘Vapor growth’ parts, it would be better to provide a simple introduction of the basic theory and procedure in the beginning.

Response: Thanks for the suggestion. We supplemented the basic theory and procedures of "hot injection" and "vapor phase growth" in the beginning.

The hot injection (HI) method has been previously developed to synthesize cadmium chalcogenides. The organometallic reagent was rapidly injected into a thermally coordinated solvent to generate discrete uniform nucleation.[38] Hot injection leads to instantaneous nucleation, quenched by the rapid cooling of the reaction mixture, as supersaturation is relieved by a nucleation explosion. At lower temperatures, this causes further growth of the existing nuclei into mature NCs without causing new nucleation.[39] In recent years, the HI method has been used to synthesize perovskite NCs. Protesescu et al. fabricated monodisperse CsPbX3 NCs with cubic shape for the first time by the HI method.[37] Typically, Cs2CO3 was added to a mixture of octadecene (ODE) and oleic acid (OA) and heated under nitrogen to form a Cs precursor. Subsequently, the cesium precursor was injected into a mixture of PbX2, ODE, oleylamine (OAm), and OA at high temperatures (150 - 200°C) (Figure 1c). The reaction system was then quenched in an ice bath to obtain NCs of different sizes, which could be adjusted by controlling the reaction conditions. Notably, the obtained NCs with different compositions displayed tunable emission spectra from 410 nm to 700 nm (Figure 1d).[29,37]

[39]de Mello Donegá, C.; Liljeroth, P.; Vanmaekelbergh, D. Physicochemical evaluation of the hotinjection method, a synthesis route for monodisperse nanocrystals. Small 2005, 1, 1152-1162.

Vapor phase growth converts the crystalline material into a vapor phase through sublimation, evaporation, sputtering, or decomposition, and then deposits it under appropriate conditions to realize the controllable atomic transfer of substances from the source material to the solid film.[43] Compared to hot injection methods, the advantage of vapor phase growth techniques is that the grown NWs tend to be of higher quality and lower defect density. Vapor growth is typically performed in chemical vapor deposition (CVD) tubular reactors, which provide a controlled and scalable method for growing high-quality semiconductors.[44,45] CVD has been widely used to fabricate halide perovskite NWs with excellent properties.[46-48]

[43] Guesnay, Q.; Sahli, F.; Ballif, C.; Jeangros, Q. Vapor deposition of metal halide perovskite thin films: process control strategies to shape layer properties. APL Mater. 2021, 9, 100703.

  1. The author mention that ‘Considering the optoelectronic properties, NWs grew in the vapor phase generally have higher quality and lower defect density’. Is this compared to the ‘hot injection’ method? More detailed explanations are needed.

Response: We thank the reviewer’s suggestion. We have explained this in detail and the corresponding content at the beginning of section “2.2. Vapor Growth”.

Vapor phase growth converts a crystalline material into a vapor phase through sublimation, evaporation, sputtering, or decomposition, and then deposits it under appropriate conditions to realize the controllable atomic transfer of substances from the source material to the solid film.[43] Compared to hot injection methods, the advantage of vapor phase growth techniques is that the grown NWs tend to be of higher quality and lower defect density. Vapor growth is typically performed in chemical vapor deposition (CVD) tubular reactors, which provide a controlled and scalable method for growing high-quality semiconductors.[44,45] CVD has been widely used to fabricate halide perovskite NWs with excellent properties.[46-48]

[43] Guesnay, Q.; Sahli, F.; Ballif, C.; Jeangros, Q. Vapor deposition of metal halide perovskite thin films: process control strategies to shape layer properties. APL Mater. 2021, 9, 100703.

  1. The authors can discuss current challenges in the application of perovskite NWs on lasers, PDs, and solar cells at the end of each part or in the Summary and Perspective.

Response: We have added the challenges in the application of perovskite NWs on lasers, PDs, and solar cells in the Summary and Perspective.

Although significant progress has been made in the synthesis and application of halide perovskites, there is still room for improvement in the device performance. For lasers and PDs, the performance of organic-inorganic perovskite NW devices is better than that of all-inorganic perovskite NW devices, but their stability is poor. Hence, high-performance all-inorganic perovskite NW devices still have great potential. In addition, the performance of NW-based solar cells is inferior to that of thin-film ones. There is an urgent need to develop excellent NW synthesis strategies for obtaining large-area and highly ordered NW arrays to improve the performance and environmental stability of solar cells. Therefore, there is still much research on the controllable synthesis and application of perovskite NWs to meet the commercial needs.

  1. The author mention that ‘the low-dimensional MAPbI3 is expected to outperform the three-dimensional structure, improving the hole migration from perovskite to hole transport layer (HTL) and separation at HTL/perovskite interface’. Would this be applicable for QDs or just limited to NWs?

Response: We thank the reviewer for the valuable comment. The improved carrier separation and migration is ascribed to their highly crystalline 1D structure, which facilitates carrier confinement in the other two dimensions. Simultaneously, few ionic defects and grain boundaries of 1D perovskite NWs contribute to efficient carrier separation and migration. (Nano Lett. 2015, 15, 2120−2126; Nano Energy 2018, 51, 192−198.) For perovskite QDs, the insulating ligands have a negative impact on carrier separation and migration, thus causing performance loss of perovskite solar cells. (J. Energy Chem. 2022, 69, 626–648) Therefore, we think this is just limited to NWs.

Reviewer 3 Report

The authors summarized the recent development of metal halide perovskite nanowires in photoelectronics, including the synthesis methods and the applications. The whole manuscript was organized in a reasonable way and a good volume of literatures has been included, although the outlook/perspectives can be far improved. Also, the authors need to carefully check the grammar and typos and make corrections accordingly. Except these, there are also some scientific items which need to be discussed and make them clearer.  

1.     First paragraph in “Introduction”, “… perovskite materials with different morphology, such as quantum dots, …” where the term “morphology” cannot include all the contents mentioned by the authors. Quantum dots, nanowires, nanoplates and thin films can be structure/shape and the morphology is more related with the film.  

2.     First paragraph in “2. Synthesis of Perovskite Nanowire ”, line 6 “…which leads NCs further…” should be “…which leads to NC further…”.

3.     In Figure 1, only d) comes from ref. [35]. How about a) – c)? if they are also adopted from the literatures, please make it clear.

4.     Second paragraph in “2.1 Hot Injection”, “Interestingly, high temperatures and long reaction times could…” should be “Interestingly, high temperature and long reaction time could…”

5.     The figure solutions are quite poor. There are words which cannot be recognised, including Figure 1c, Figure 3a, Figure 4e,4f and 4j, Figure 8a, Figure 9a, Figure 10 and 11.

6.     In section 2.3.3, line 5 in first paragraph, “leading to transform CsPbI3” should be “leading to the transform of CsPbI3…”   

7.     Please check the grammar of the caption of Figure 8b.

8.     Please check “Under light excitation, The Fabry-Perot maser occurred in CsPbBr3 NWs with a starting wavelength of 5 μJ cm-2, and the maximum quality factor of the NW cavity was 1009 ± 5 (Figure 10f-l).” “The Fabry-Perot” should be “the”. 5 μJ cm-2 is not the wavelength and it should onset of lasing. “Figure 10f-I” should be “Figure 10f-i”.

9.     In photodetector parts, the author used “responsiveness” which should be “responsivity”. “detection rate” should be “detectivity”.

Author Response

Response to Reviewer

The authors summarized the recent development of metal halide perovskite nanowires in photoelectronics, including the synthesis methods and the applications. The whole manuscript was organized in a reasonable way and a good volume of literatures has been included, although the outlook/perspectives can be far improved. Also, the authors need to carefully check the grammar and typos and make corrections accordingly. Except these, there are also some scientific items which need to be discussed and make them clearer.

Response: We thank the reviewer for the detailed suggestions to improve the quality of our manuscript. We have addressed the issues point-to-point in the revised manuscript. Our responses to the reviewer’s comments are listed as followings.

  1. First paragraph in “Introduction”, “… perovskite materials with different morphology, such as quantum dots, …” where the term “morphology” cannot include all the contents mentioned by the authors. Quantum dots, nanowires, nanoplates and thin films can be structure/shape and the morphology is more related with the film.

Response: Thanks for indicating our mistake. We have changed “morphology” to “structures”.

  1. First paragraph in “2. Synthesis of Perovskite Nanowire”, line 6 “…which leads NCs further…” should be “…which leads to NC further…”.

Response: Thanks for indicating our mistake. We have changed “…which leads NCs further…” to “…which leads to further growth of NCs…”.

  1. In Figure 1, only d) comes from ref. [35]. How about a) – c)? if they are also adopted from the literatures, please make it clear.

Response: We thank the reviewer for the valuable comment. In Figure 1, a) – c) are the pictures drawn with PowerPoint by ourselves.

  1. Second paragraph in “2.1 Hot Injection”, “Interestingly, high temperatures and long reaction times could…” should be “Interestingly, high temperature and long reaction time could…”

Response: Thanks for indicating our mistake. We have changed “Interestingly, high temperatures and long reaction times could…” to “Interestingly, high temperature and long reaction time could…”.

  1. The figure solutions are quite poor. There are words which cannot be recognised, including Figure 1c, Figure 3a, Figure 4e,4f and 4j, Figure 8a, Figure 9a, Figure 10 and 11.

Response: We thank the reviewer for the valuable comment. We have replaced these pictures with high-resolution ones.

  1. In section 2.3.3, line 5 in first paragraph, “leading to transform CsPbI3” should be “leading to the transform of CsPbI3…”

Response: Thanks for indicating our mistake. We have changed “leading to transform CsPbI3” to “leading to the transformation of CsPbI3…”.

  1. Please check the grammar of the caption of Figure 8b.

Response: Thanks for indicating our mistake. We have replaced the caption of Figure 8b to “Proposed growth process of CsPbBr3 NCs obtained without and with precursor dissolving.”.

  1. Please check “Under light excitation, The Fabry-Perot maser occurred in CsPbBr3 NWs with a starting wavelength of 5 μJ cm-2, and the maximum quality factor of the NW cavity was 1009 ± 5 (Figure 10f-l).” “The Fabry-Perot” should be “the”. 5 μJ cm-2 is not the wavelength and it should onset of lasing. “Figure 10f-I” should be “Figure 10f-i”.

Response: Thanks for indicating our mistake. We have modified the sentence to “Under light excitation, Fabry-Perot lasing occurred in the CsPbBr3 NWs with an onset of 5 μJ cm-2, and the maximum quality factor of the NW cavity was 1009 ± 5 (Figure 10f-l).” For the issue “(Figure 10f-l)”, we have carefully checked the letter “l” both in the caption and figures, which is correct.

  1. In photodetector parts, the author used “responsiveness” which should be “responsivity”. “detection rate” should be “detectivity”.

Response: Thanks for indicating our mistake. We have changed “responsiveness” and “detection rate” to “responsivity” and “detectivity”, respectively.

Round 2

Reviewer 3 Report

The authors checked the whole text and did corresponding corrections. I agree to publish.